# Ecological changes over 90 years at Low Isles on the Great Barrier Reef

Maoz Fine [1,2]*, Ove Hoegh-Guldberg [3,4], Efrat Meroz-Fine [2] & Sophie Dove[3,4]

Coral reefs are under increasing stress from local and global factors. Long-term perspectives are becoming increasingly important for understanding ecosystem responses. Here, we provide insights from a 91-year study of the Low Isles on the northern Great Barrier Reef (GBR) that begins with the pioneering Great Barrier Reef Expedition (1928-29). We show that intertidal communities have experienced major phase-shifts since 1928, with few signs of a return to the initial state. Coral communities demolished by cyclones 50 years ago and exposed to multiple stressors have yet to recover. Richness and diversity of these communities systematically declined for corals and other invertebrates. Specifically, massive corals have replaced branching corals, and soft corals have become much more numerous. The long-term perspective of this study illustrates the importance of considering multiple factors in reef decline, and potential recovery, of coral reefs, and the importance of tracking changes in community structure as well as coral abundance over long periods.

[1] The Mina and Everard Goodman Faculty of Life Sciences, Bar-Ilan University, 52900 Ramat-Gan, Israel. [2] The Interuniversity Institute for Marine Science, P. O. Box 4698810300 Eilat, Israel. [3] School of Biological Sciences, The University of Queensland, St. Lucia, QLD 4072, Australia. [4] Global Change Institute, University of Queensland, St. Lucia, QLD 4072, Australia. *email: maoz.fine@biu.ac.il

Coral reefs are under rapid decline, putting the food and livelihoods of hundreds of millions of dependent people at risk. Intensifying climate change has pushed many coral reefs close to collapse over the last few decades, with their ability to recover becoming increasingly uncertain[1]. Rising temperature anomalies and associated increases in mass coral bleaching and mortality events have spurred research efforts to understand and project how coral reefs might change in an increasingly warm and acidic ocean[2], particularly for near coastal coral reefs, which also suffer other stressors at varying spatial scales, frequencies and intensities[3,4].

Disturbances play an important role in structuring coral reef community composition[5,6]. While some coral reef assemblages may be resilient to large-scale natural disturbances[7,8], an increasing frequency of global and local disturbances make it difficult to assess if recovery leans towards a pre-disturbance composition[9] or undergoes a phase shift to a new steady state[10,11]. A shift in reef-building coral composition and structure often affects the many reef-associated fish and invertebrates species[12], which in turn compromises ecosystem resilience through reduced functional redundancy[13] and ecological feedback loops[14]. It is essential to identify long-term trends in reefs community dynamics in the face of the ongoing changes[15], if we are to effectively manage coral reefs into the future[16]. The realisation of environmental change has driven large increases in the monitoring of coral reefs, mostly beginning 2–4 decades ago. Longer-term, repeatable ecological data beyond 40 years are rare[17], especially data sets for subtidal reef sections, largely due to the absence of scientific diving techniques prior to the late 1940s.

Here, we use an unusual opportunity to understand the survival of coral reefs in the context of close to a century of biological data. The Great Barrier Reef Committee and the Royal Society of London sent an expedition to study Australia's Great Barrier Reef (GBR) in 1928[18]. Members of the GBR expedition (GBRE) lived on Low Isles (LI), an inshore coral island (off Port-Douglas, Queensland) for over a year. During this time, the GBRE documented the environmental conditions surrounding the coral reefs of the LI as well as the community structure of tidal and subtidal communities[19], the latter using a diving helmet. This pioneering study was the first in a series of highly focused expeditions to the LI, which have explored its geology, prehistory and ecology (Fig. 1a). Another innovation associated with the legacy of the 1928 expedition was an accurate, aerial photography-based mapping of the island[20,21]. This enabled the 1928 GBRE to map the intertidal habitats at LI as well as three 100–400 by 0.9 m belt transects (T1–3), which were accessed by wading or diving at different parts of the island (Fig. 1b). Using the 1928 expedition's highly accurate mapping of the reef, we were able to (1) revisit and sample precisely the same intertidal locations explored by the GBRE and a follow-up study in 1954[22]; (2) repeat three permanent traverses 76, 87 and 91 years later, and thereby form the longest ecological survey of coral reefs to date. These data sets enabled us to extract a high-precision account of the ecological changes to intertidal and subtidal coral reef habitats over this long time. Our study reveals a long-term systematic decline in coral and invertebrate richness at the LI reef since the GBRE time and a phase-shift to a low complexity coral community.

## Results

### Changes in environmental conditions.
The environmental conditions around LI have changed significantly since 1928–29. Average annual Sea surface temperature in the northern section of the GBR is about 0.7 °C higher now than it was in 1928[23] and winter temperature in the Anchorage at LI is significantly higher during the last decade compared with temperatures measured twice a day in 1928–29 (Fig. 1c). In 1928, Orr and Moorhouse[24] concluded that "it is unlikely that the temperature in the Anchorage or elsewhere near LI ever reaches the lethal temperatures for corals", highlighting the importance of long-term records. Since that time, mass coral bleaching events have affected coral reefs within the area. Seawater pH is lower by about 0.1 pH units and flood episodes occur much more often, with increasing amounts of sedimentation and nutrients flowing out onto coastal areas[25,26]. Sea level is now higher by over 20 cm than it was in 1928. Cyclones, however, have shaped the shallow reefs more than any other environmental factor and for a longer period[3] (Fig. 2).

### Intertidal and subtidal communities.
Our surveys of the intertidal sites described by the GBRE and those of a later 1954 GBRE[22] suggest that coral cover has declined along with a major decline in species richness. Intertidal habitats named by the GBRE as the Porites pond, Montipora Lawn, Fungia Moat and Madrepore moat, reflecting the dominant reef-building coral fauna in these areas in 1928–29, are now mostly devoid of all corals (Supplementary Table 1).

Subtidal coral communities, in contrast to intertidal coral communities, did not show a consistent decline over the past century (Fig. 3a). In 1928-29, subtidal hard coral cover ranged from 8% in the Anchorage traverse (T2) to 22% in T1 and 42% in T3. Based on Manta-tow and permanent transect surveys, AIMS LTMP[27] reported LI coral cover (1986 to 1998) was in the range of 21–40%. A mass coral bleaching event in 1998, as well as a period of elevated numbers of Crown of Thorns Starfish (CoTS), and a cyclone, in early 1999, resulted in the area covered by hard corals dropping to <10%. In 2004, we found that coral cover was 22%, 18% and 16% in T1, T2 and T3, respectively. AIMS LTMP reported a reef-wide median of manta board surveys of 30–40% coral cover in 2010. Coral cover in 2015 (present study) was high in T3 (53%), but relatively low in T1 (18%) and T2 (10%), and was probably due to a tropical storm hitting the lee side of the island in 2014. In 2017, estimated coral cover in scattered locations (and single datum per location) around LI, ranged from zero to >70%[28]. Coral cover was low in all three traverses in March 2019, being 17%, 15% and 22% for T1, T2 and T3, respectively.

While coral cover oscillated with the numerous stressors, species richness in the three traverses has consistently declined over time (Fig. 3b–d). The ratio of massive to branching corals was much greater in 2019 than in 1928 (Fig. 3e and Supplementary Fig. 1) as was the ratio of soft to hard corals (Fig. 3f). The Anchorage, in particular, shifted from scleractinian (hard coral) to soft coral dominated reef over the 90 years of this study. In 2004, there were over fivefold the number of soft coral colonies in the Anchorage than in 1928. This changed only little in 2015 and 2019 (Fig. 3f). The most dominant space occupiers in the Anchorage were soft corals from the genera Sarcophyton, Lobophytum and Sinularia.

Coral colony size has also changed considerably from 1928. Size range of Acroporid corals is 30% smaller in 2004–2019 than the size range in 1928–29. Similarly, the diameter of massive corals was 15% lower. Revisiting 13 intertidal sites from the 1954 survey[22] (Supplementary Fig. 1) also revealed a major decline in coral species richness and near extinction of numerous species (Supplementary Table 2). Four sites that had 13–26 species, some of which very dominant in 1954, now have none. Apart from corals, invertebrates that were reported to be common in these sites were no longer present, despite extensive searches across a substantial area of intertidal reef (Supplementary Table 2).

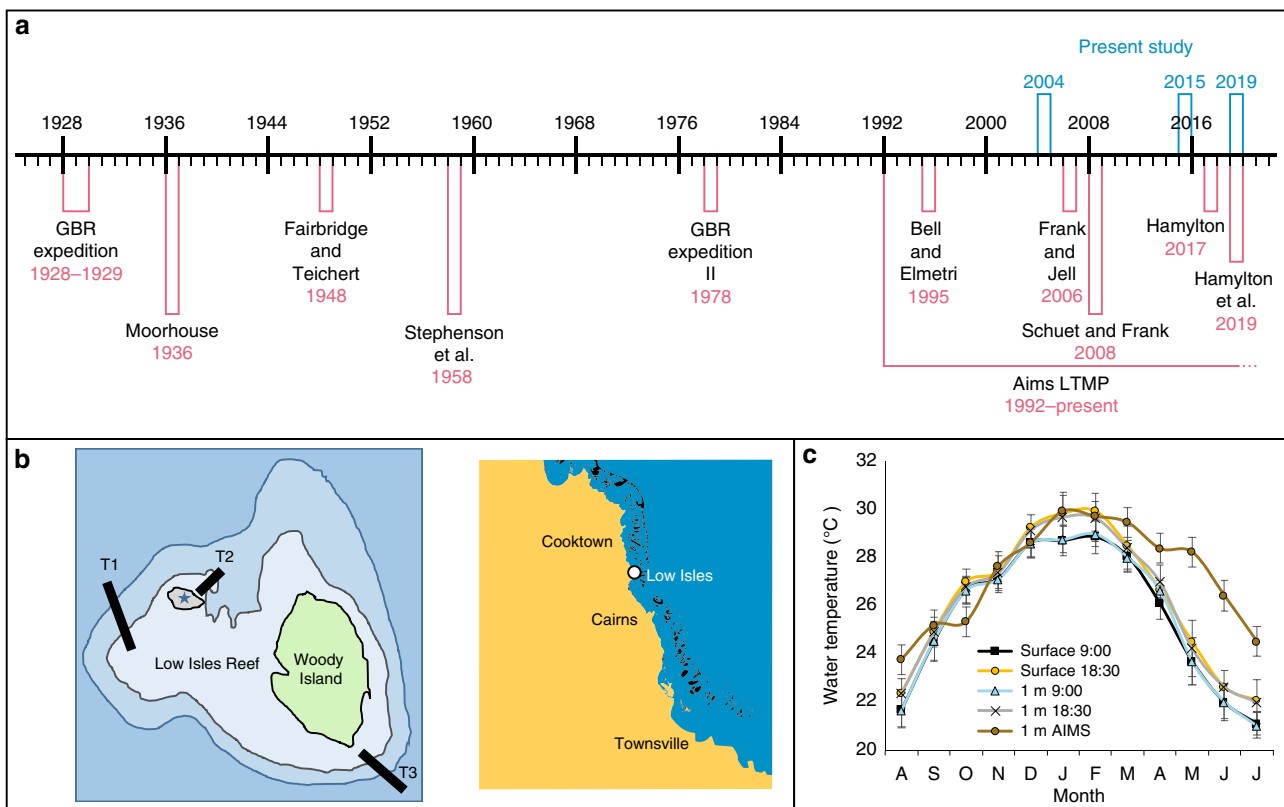

**Fig. 1** The historical background of the Low Isles reefs. **a** A timeline of key studies[20,22,26–30,50,53–55] at the Low Isles historic reef since 1928 to the present study (years 2004, 2015 and 2019). **b** A map depicting the location of Low Isles in the northern sector of the GBR (inset) and the three traverses performed along a depth gradient, by the GBRE[19]. A star represents the LI lighthouse. **c** Seawater temperature as measured twice a day by the GBRE in the Anchorage at the surface and at 1 m depth from August 1928 to July 1929, and a record of temperature data from an Australian Institute of Marine Science (AIMS) data logger on the LI reef flat at 1 m depth (monthly average 1996–2016)

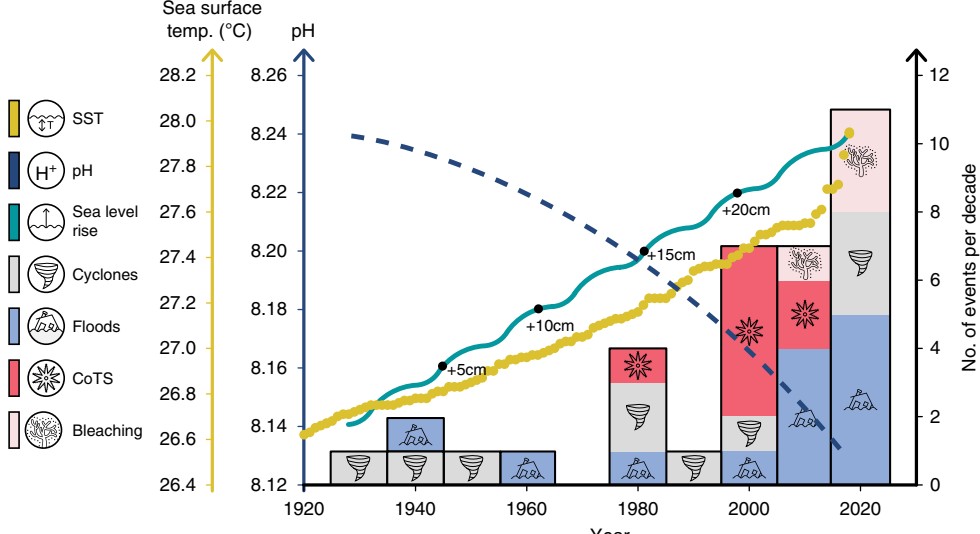

**Fig. 2** A schematic of changes in environmental conditions over time at Low Isles. Temperature and sea level rise and pH decrease (lines) and stacked bars showing accumulation of stress (Crown of Thorns Starfish (CoTS), runoff, cyclones and bleaching) events per decade, highlighting the increased pressure on reefs of the area (see Supplementary Table 3 for details of stress events). While the temperature curve is based on data for the Coral Sea, sea level and pH are illustrative of start and end point (1928–present)

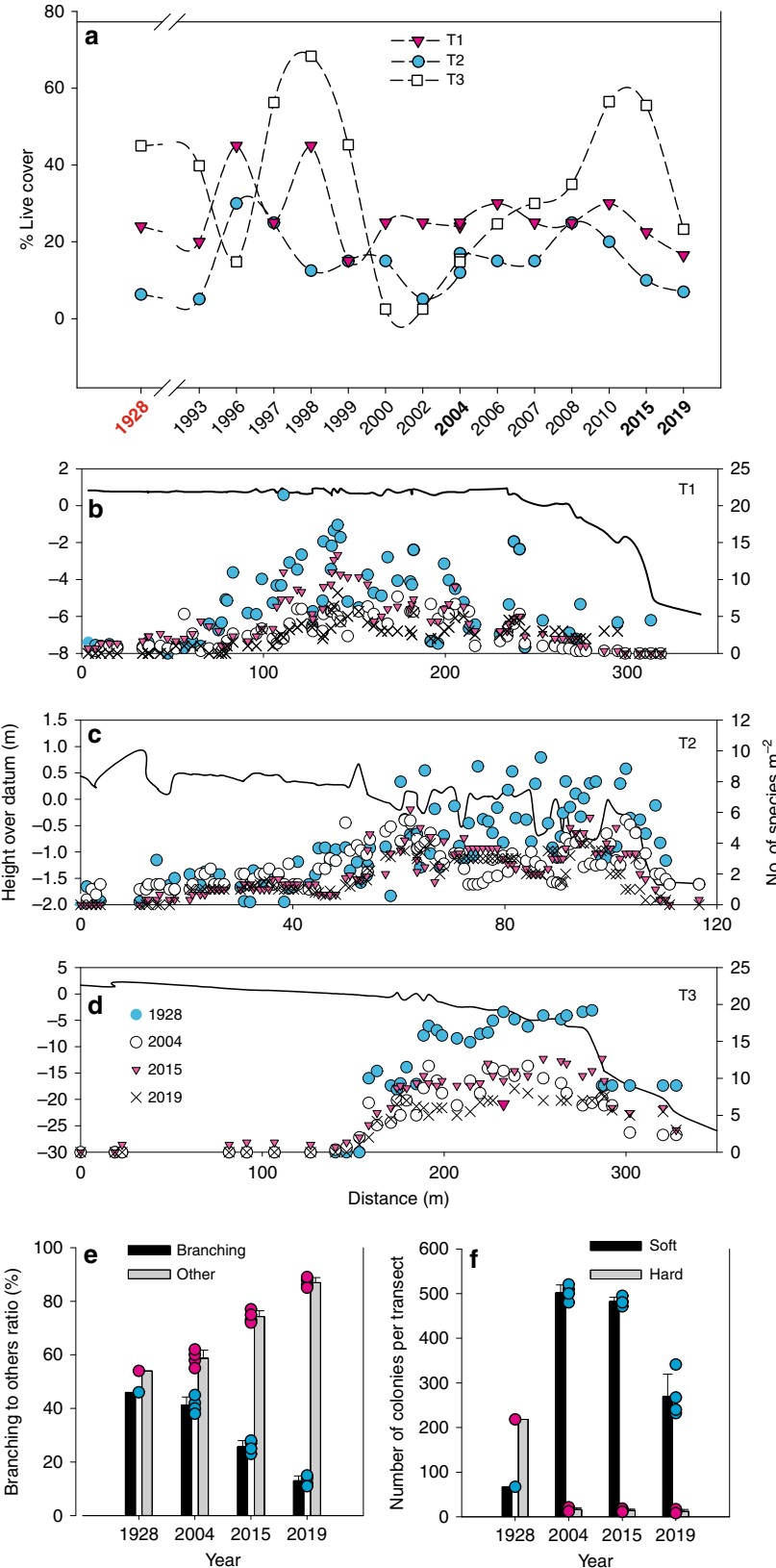

Comparing our data (i.e. years 2004, 2015, 2019) with that of the 1954 study, we find a decline in the reported species in all 13 intertidal study sites—from a total of 40 species and a Shannon–Winer index (S–W) of 2.68 in the moats in 1954 to 21 species and a S–W of 1.23 in 2004. Subsequent study over the following period revealed the same trend: 18 species and an S–W

of 1.12 in 2015, and 10 species and an S–W of 1.1 in 2019 (Supplementary Table 2).

Carefully comparing the eight reef sections drawn underwater (true to scale), the present study found significantly lower diversity at the same locations (Fig. 4a–d). These selected locations, which were described by the GBRE as representative

**Fig. 3** Changes in coral communities in the Low Isles from 1928 to 2019. **a** Live hard coral cover in traverse T1–3, data from the 1928 expedition and from the present study for 2004, 2015 and 2019. Data for other years were extracted from the AIMS LTMP reports from Manta tow surveys performed near the original traverses (please note, distance between sampling points is inconsistent). Coral species richness in each Traverse (**b**–**d**) is presented along a depth contour (continuous line) and symbols stand for number of species identified per square metre along the Travers in 1928 (turquoise circle), 2004 (black circle), 2015 (magenta triangle) and 2019 (black x). Datum refers to water level at lowest low tide[19]. Changes in coral growth form (**e**) across all traverses show a shift, within hard coral, from a high percentage of branching species (turquoise circles) to dominance of massive and encrusting species (magenta circles) and in Traverse 2 (Anchorage) a shift in dominance from hard (magenta circles) to soft (turquoise circles) corals (**f**). Bars represent average ± s.d.

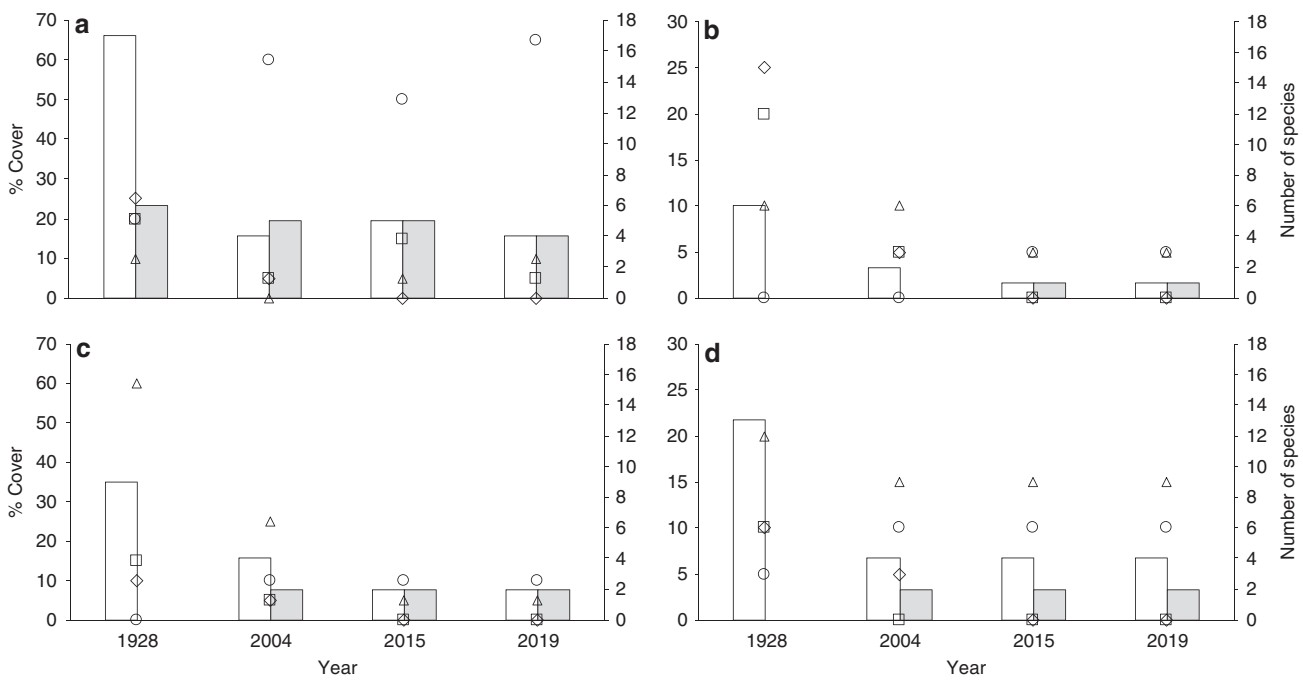

**Fig. 4** Data from selected representative reef sections, which were drawn by the GBRE in 1928 (true to scale)[19] and revisited and photographed in 2004, 2015 and 2019, demonstrating the changes in coral percent live cover and number of species since 1928. **a** Coral Head situated off the northeast corner of LI. **b** An area in the Madrepore Moat. **c** An area in the Western Moat where coral growth was at its maximum in 1928. **d** An area on the seaward slope of Traverse 1 where coral growth was at its maximum in 1928, dominated by *Acropora* species. Markers represent percentage live cover of branching (diamond), massive (triangle), encrusting (square) and soft (circle) corals. Columns represent number of hard (blank) and soft (grey) corals

of the reef section, have all shifted to much lower levels of diversity and complexity.

## Discussion

LI, one of the most studied coral islands in the world[29], beginning with the pioneering studies by the 1928 GBRE, has undergone dramatic geomorphological[28] and ecological change. Since the time of 1928 GBRE, the coral reefs of the LI have become consistently degraded, with changes starting as early as 1934, when a powerful cyclone hit the area[30]. Yet, in 1954, 4 years after another strong cyclone (1950), reefs still had relatively high coral cover and diversity[22]. Modification of the ramparts[28] that previously dammed water over the corals during low tide may explain their disappearance, although the increased sea level (~20 cm since 1928) might have been expected to counter this as corals repopulated reef-flat habitats over time[31]. Coral cover strongly predicts the capacity of reefs to track sea level rise (SLR). Therefore, for the LI, under the current low coral cover, SLR might result in reef drowning[31], depending on the rate of SLR, as modelled for Lizard Island[32].

Cyclones are often selective in shaping coral communities, removing first branching growth forms followed by small massive corals[3]. Larger massive corals as well as soft corals are more resilient to cyclone impacts as recorded after cyclone Rona (February 1999)[33]. By contrast, branching species are not only more fragile, but often settle on coral rubble[22] arising from previous cyclones. Unconsolidated coral rubble is highly unstable under wave action, potentially inhibiting the survival of recruits. Encrusting growth forms, on the other hand, showed a faster rate of recovery following Rona[33]. Branching coral species are also more sensitive to environmental change, including temperature stress. This may explain the near local extinction of many species of branching corals at the LI, which were wiped-out physically (cyclones) from the LI and were unable to recruit back due to repetitive floods and bleaching events from marine heat waves[5,15]. Furthermore, cyclones have relocated reef flat structures and infilled some with sediments, making them inhospitable for corals[28].

Lower coral cover and structural complexity in the intertidal habitats was accompanied by reduced diversity of invertebrates (Supplementary Table 2). Over half of coral-associated invertebrates have an obligate association with live corals[34]. Some invertebrates are highly selective for specific coral species that provide food or habitat. The progressive loss of corals from tropical reefs has been reported to result in the substantial loss of invertebrate diversity[35].

Predicting recovery of reefs following environmental stress is challenged by the type, frequency and amplitude of events (disturbance regime)[7], supply of propagules for recovery (connectivity) life history of dominant corals[36,37] and presence/

absence of critical functional groups[38]. It may also vary between habitats (depth, reef zone, orientation)[17]. Multiple acute and chronic disturbances may hamper community recovery, as observed in many reef systems[39–41]. Recovery in abundance does not necessarily imply that the assemblage has recovered (species diversity, size frequency, fecundity and growth)[7]. Succession following disturbance begins with pioneering species occupying space. Only later, with intermediate disturbances[42], other species are able to penetrate the pioneering community towards build-up of a rich climax community. In the case of the LI, following disturbances, pioneering species tend to occupy available space but are not joined by other species, or at least very slowly. Understanding succession of a coral community, from disturbance, to a climax phase[17] and whether the community has reached its pre-disturbance state is of great importance.

The most common proxy to post-disturbance coral reef recovery is live coral cover[43,44], although a focus on the abundance of coral, measured as percent live cover, is potentially misleading[45,46]. For example, while coral cover at LI oscillated with the numerous stressors, species richness has consistently declined over time (Fig. 3b–d). These observations are in agreement with other studies demonstrating that the coral reefs at the LI recover but fail to reassemble, lagging 21 years behind coral recovery[45]. Highest number of soft and hard coral species per square metre along the traverses in 1928–29 was 25, 10 and 20 for T1, T2 and T3, respectively. Species richness was 50% lower in 2004 and 2015 when compared to 1950 and 1928. In 2019, species richness has dropped further to 37% of the original measurements in 1928. Understanding the timing of these changes to coral abundance and species richness has the potential to identify causal factors. Recovery of an isolated reef in western Australia following severe bleaching event continued for about 12 years with remaining colonies increasing in size and adding to coral cover before propagules supply resumed, resulting in recruitment of new colonies[47]. Similarly, coral communities in four different Palauan reef habitats reached a climax community 9–12 years after the 1998 bleaching event[17]. The survey conducted in the moat areas of the LI in 1954[22], for example, already had lower coral cover compared with 1928–29 as a result of the 1950 cyclone, but still had similar species compositions. In the latter case, this may be due to higher soft coral recovery rates following physical disturbance, as well as, the preference of soft corals for increased particulate organic matter in the surrounding waters. These changes may reflect repetitive decimation of primarily branching coral species and lower levels of regrowth during shorter recovery periods. It may also reflect reduced growth rate of corals due to elevated temperature and ocean acidification[18], although substantiating this from the current data set is not possible.

Many aspects of the coral reefs of the LI have changed over the past 90 years of study, with substantial visual differences and little evidence of adaptation. The capacity of a reef to absorb disturbance without shifting to an alternative state depends on factors such as the pre-disturbance community, the type and frequency of the disturbance and whether it has stabilised, and the interplay between local and global stressors[48]. Chronic stress may select for species that are more resistant to disturbances, which in turn will enable recovery of coral cover[45] but many Inshore reefs on the GBR already dominated by relatively resistant, slow-growing coral and soft corals now take longer to recover following disturbances[45]. Reef communities that slowly regain coral cover but fail to reassemble a diverse community are more sensitive to further change as can be seen at the LI. The degree to which processes at LI are representative of other inshore reefs or the entire region is unclear due to the lack of other long-term records. Jonker et al.[16] reported that turbid inshore reefs

such as the LI had the most dissimilar coral assemblages compared with offshore ones although richness and diversity of juvenile coral assemblages were comparable to those further offshore. That said, reports from the last two decades[3] suggest that inshore and offshore reefs show a similar trend of decline. These changes in the inshore areas of the world's largest continuous coral reef form the narrative for coral reefs globally, illustrating the key role of multiple cumulative stresses and unparalleled change over long periods of time.

## Methods

**Locating and georeferencing historic sites.** LI Reef is an inshore reef located in the northern GBR, Australia (16°23′ S, 145°34′ E). To track the footsteps of the coral ecology work performed by the 1928 expedition and a follow-up expedition in 1954, the original GBRE map of LI was used[20,21,30,49–51]. To correlate a global positioning system (GPS) with the original maps, we digitised the maps using Georeferencer (Klokan Technologies GmbH) and Google Earth. The lighthouse and the base of an old jetty were used as georeferenced sites. By linking points on the image with those same locations in the geographically referenced data, a polynomial transformation is created, converting the location of the entire image to the correct geographic location. The location of habitats explored and reported by the 1928 GBRE and 1954 expedition were mined from the georeferenced map and a handheld GPS (Garmin 2004, 2015; Suunto Ambit3 in 2019) was used to return to these locations. The general description of the moats, ponds and intertidal sections was performed by four people (M.F., E.M.-F., S.D. and O.H.-G. in 2004, M.F. and O.H.-G. in 2015, 2019) who overlapped and repeated their visits to each site. The general description was then integrated from these surveys and compared with older reports[19,22].

**Survey of historic traverses and habitats.** The three traverses measured by the GBRE[19] were used as permanent transects. Traverse 1 on the western side, Traverse 2 on the north western and Travers 3 in the south-eastern side of the island (Fig. 1b). Intertidal sections of the traverse were performed at low tide in 2004 and 2015, at high tide in 2019. Subtidal sections were performed using a diving helmet in 1928–29 and SCUBA from 2004 onwards. In 1928–29, a rectangular frame (1 × 2 yards and cross-partitioned into square feet)[19] was placed successively along the travers and all corals and algae were recorded. Ground glass was used to write or draw on, with a pencil. Similarly, subtidal surveys in the present study were performed at high tide by SCUBA diving. Traverses were measured using a measuring tape and a digital camera. In each site, three traverses were performed in parallel (~10 m apart) to cover well the area of the 1928 traverse from start to endpoint. In 2004, digital photos 1 × 1.5 m with an overlap of at least 60 cm were stitched (Adobe Photoshop) to form a continuous panoramic traverse on which a set of 1 × 1 m quadrate (true to scale) was electronically overlayed. Analyses of coral cover, species composition and size were drawn from the quadrates.

A Nikon AW-130 underwater digital camera was used for the traverses during the 2015 and 2019 expeditions. The measuring tape was videotaped along with still image being taken at the beginning of a transect (to record the GPS position using the camera). A still image was taken at the surface before the divers submerged during the measurement of the subtidal sections. The camera recorded also the depth-contour, which was later compared with the 1928 depth profile. A second still photo was taken at the end of each transect at the surface. CPCE[52] was used to analyse coral species diversity, live cover and colony size. Nine regularly spread data points were overlaid on every 1 m of the transect and the coral/substrate underneath every point was recorded. The data were compared with the data mined from the GBRE 1928–29 reports using WebPlot Digitizer V.4.1.

The shallow water historic habitats from 1928 and 1954 were surveyed by a member of the team, beginning in the point location and covering an area of ~80 m² (5 m radius) around the starting point. Coral species and sizes were recorded as well as other invertebrates and algae. A general description of each habitat/site was written, including substrate type, landscape and dominant organisms. Photos were taken of corals, invertebrates, algae and substrate. This data were then compared with the historic records for the same locations.

**Comparison of historic and recent reef images.** Specific detailed illustrations of selected areas along the traverses, originally drawn underwater on ground glass in 1928, were analysed. Using the 1928 traverse bathymetric profile and distance along the traverse to locate the drawn historic sections, we compared with the 2004, 2015 and 2019 photographs of the same location. CPCE[22] was used on both drawings and images to extract coral cover and species composition.

**Additional data sources.** The AIMS long-term monitoring program[27] was used as a source of coral abundance and community structure. The AIMS manta tow surveys are done at similar depths and proximity to this study traverses, and as such serve a data source for comparison. It should be noted, however, that these surveys are performed longshore whereas the GBRE traverses, and therefore ours, were performed perpendicular to the shore, along a depth contour.

**Permission to carry out fieldwork**. Fieldwork was carried out under Great Barrier Reef Marine Park Authority and Queensland Parks and Wildlife Service Permit G18/4.261.1

**Reporting summary**. Further information on research design is available in the Nature Research Reporting Summary linked to this article.

## Data availability

Survey Images will be publicly available on the CoralNet https://coralnet.ucsd.edu/, and data on reef locations, coral live cover and diversity will be available upon request from the authors.

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

## Acknowledgements
We thank Fiona N. Hoegh-Guldberg, Yam Fine, Chrisopher O. Hoegh-Guldberg and Dan Fine for field assistance; S. Green and D. Harris for organising field trips; the Low Isles caretakers (2004–2019) for their kind assistance, and Quicksilver and Sailaway for their support during field trips.

## Author contribution
M.F., O.H.G. E.M.F. and S.D., conceived the ideas and designed methodology, collected the data and analysed the data; M.F. and O.H.G. led the writing of the manuscript. All authors contributed critically to the drafts and gave final approval for publication.

## Competing interests
The authors declare no competing interests.
