## [Peer Review File · Nature Communications]

Reviewers' comments:

Reviewer #1 (Remarks to the Author):

Fine et al is a sound and novel assessment of a long-term ecological data set on a coral reef- indeed the longest such data set that there is. The paper will be useful to scientists in the field for its consideration of such a long-term data set, consideration of coral and reef health indicators beyond just coral cover, and comparisons between drivers of change at different temporal and spatial scales (eg water quality vs cyclones). I also like the incorporation of long-term physical and chemical information with the biological.

My only real concern relates to replication and generalization. Given the locations of the three 'traverses' they don't look like replicate samples of the same habitats at LI reefs, rather each is a 'traverse' through a different habitat within the Reef system. Of course, this isn't under the control of the authors, and I'm not expecting them to redesign surveys conducted in 1928. But I do think the paper should give consideration to what habitats were sampled and to what extent there is any replication. Does the survey design allow generalization about LI reef as a whole, or just the three specific 'traverses'? This concern is amplified when I consider the title's reference to 'the northern Great Barrier Reef'. It is debatable whether or not the three 'traverses' are representative of LI reef as a whole, but I contend they are almost certainly not representative of the northern GBR as a whole. LI reef is close to shore adjacent to a part of the catchment with substantial agricultural development. There are other northern GBR reefs in similar locations, but most of them are quite different (far offshore and/or not adjacent to intensively developed catchment). I think the title should be edited to better reflect the true geographic scope of the data presented and the text should give some consideration to the relationship between LI reef and other GBR reefs.

Overall, I think this is a great paper and I look forward to seeing it published.

David Wachenfeld

Reviewer #2 (Remarks to the Author):

Recommendation: accept with minor revision

General Comments

The authors have presented a unique set of data on a coral reef that spans 90 years. There is great concern globally with respect to how natural systems are changing. This paper will be well received as there are few base lines, especially for coral reefs, to go from. Furthermore the 1928-29 expedition collected both physical and ecological data which adds value to the long-term comparisons.

Other positive points – the data are from the intertidal and subtidal, multiple taxa were studied including corals, molluscs, echinoderms macro crustaceans and reef architecture. It is interesting that one of the 'villians' damaging reefs today, the Crown-of-thorns star fish, has disappeared. Not only was coral cover studied, but changes in species composition (e.g., the replacement of branching corals with massive corals) were documented. Again few studies have been encompassed such a long period to document such changes.

See 'Minor comment re Fig 1 – the drop in pH seems strangely linear!

Minor Comments

Line 19 – it is unclear how many cyclones have actually caused damage?

Lines 32-54 – the 'Main section/Introduction is coherent and clearly states why there would be international interest

Line 83 re reference 10 – the authors should provide some mainstream literature on the this

temperature change – certainly CSIRO has published changes from the 1960s

Line 90 – another good reference on runoff and nutrient levels is Furnas (2003) 'Catchments and Corals: terrestrial runoff on the GBR

Line 91 – the measured rise in sea level is interesting. You would expect the intertidal to be more protected with more water, but that does not appear to be the case? Are there data on the growth/erosion of shallow carbonates.

Line 112 re: 'near local extinction – of course this refers to the sampling sites – not necessarily the whole reef which may have refugia?

Line 145 the number of intertidal sites (n=10) from 1954 was convincing.

Line 148 – re loss of other invertebrates (e.g., echinoderms, molluscs crustaceans), there was no comment on why - loss of food such as live coral, destruction of reef architecture .. etc?

Line 164 Re: species of soft coral – that is a species diverse group –are any species or family names available?

Figures – the quality of figures could be improved – perhaps redrawn (e.g. Fig. 1). I suggest the addition of a present day google map would be complementary – there is space below 1d!?

Figure two – there are no error bars for 1928 – say why there was only one value in the records?

Live cover does not distinguish hard coral from soft coral? - I suggest the symbols are linked with lines as that would make the long-term trends easier to interpret.

Figure 3 it is hard to be convinced about some of the comparisons given the image quality. Perhaps larger paired image in the extended figures would be helpful.

re comparison c versus d and e versus f. How do you know this was the exact same location? GPS targeting may not be good enough for this scale.

Re: extended figure 1 – it would be good to see percentage cover by depth. This is a more common metric in the literature. Further colony number may not related to percentage cover very well – eg one colony may cover a large area while small colonies may be abundant, but cover little.

The extended tables add value to the paper.

Mike Kingsford

Reviewer #3 (Remarks to the Author):

This is a very interesting paper about long-term trends in reefs community dynamics.

The study shows that communities have experienced major phase-shifts over the last 90 years, with few signs of return to the initial state. Specifically, massive corals have replaced branching corals, and soft corals have become much more numerous than hard corals. Others invertebrates, such as grazing echinoderms, gastropods, crustaceans and many other non-coral invertebrates are also disappeared.

1) Authors consider multiple factors to explain these changes, however, they don't specify how exactly each one could explain the changes observed in the community. The intertidal zone is the

most temporally and spatially variable of all marine habitats. It is characterized by environmental extremes (temperature, salinity, desiccation, nutrient supply), yet there are also strong biotic interactions determining community structure. Therefore, organisms living in the intertidal zone have evolved a variety of anatomical, physiological and behavioral adaptations that enable them to survive in this challenging habitat. It seems to me, that recurrent storms could explain by itself most of the changes, and the fact that subtidal coral communities, in contrast to intertidal coral communities, did not show consistent decline over the past century seems to support this hypothesis, but I would like the authors to delve deeper into how other different factors considered could explain the changes registered: sea level rise, for example, must be also very important. In the time that the authors made the study, the original intertidal zone is not so intertidal, because the sea rose over 15 cm (see Fig. 1).

2) They also may consider other abiotic factors that could be responsible of the observed data, such as wave exposure and tidal height, which have been shown to have strong effects on the dynamics of intertidal communities, contributing to invertebrate abundance and species richness; all this is also related to the rise in sea level.

3) I also would like that authors explore additional hypothesis, which serves to explain the impoverished diversity, based on the unstable composition of faunas in remote marginal regions, and in the low resilience of these faunas (see references below). Due mainly to physical perturbations commented above, species already living near their tolerance limits become locally extinct and are not soon replenished after disturbances (strong cyclones) because of their isolation from source populations.

4) Apart from corals, invertebrates that were reported to be common in these sites were no longer present, despite extensive searches across a substantial area of intertidal. Could be more specific pls? Species? Genera?

5) Fig. 1: Pls define CoTS =

6) Surveys of the intertidal sites described by the GBRE and those of a later 1954 GBR expedition, suggest coral cover has declined along with a major decline in species richness. However, coverage seems to increase in T3 (Fig. 2). Coral cover in 2015 was high in T3 (53%), was low in all three traverses in March 2019. Could explain a little more why cover oscillated, and even increased, and in contrast species richness consistently declined over time?

7) Please consider these relevant references in the revision of the ms

Brown BE, Suharsono. Damage and Recovery of Coral Reefs Affected by El Nino Related Seawater Warming in the Thousand Islands, Indonesia. *Coral Reefs*. 1990; 8, 163-170.
<http://dx.doi.org/10.1007/BF00265007>

Russ G, Alcalá A. Natural fishing experiments in marine reserves 1983–1993: roles of life history and fishing intensity in family responses. *Coral Reefs*. 1998; 17: 399-416.
<https://doi.org/10.1007/s003380050146>

Connell J. Disturbance and recovery of coral assemblages. *Coral Reefs*. 1997; 16(Suppl 1): S101-113. <https://doi.org/10.1007/s003380050246>

Dollar SJ, Tribble GW. Recurrent storm disturbance and recovery: a long-term study of coral communities in Hawaii. *Coral Reefs*. 1993; 12: 223-233. <https://doi.org/10.1007/BF00334481>

González-Cabello A, Bellwood DR. Local ecological impacts of regional biodiversity on reef fish assemblages. *Journal of biogeography* 2009; 36 (6): 1129-1137.

11.7.2019

We thank the reviewers for acknowledging the importance of our study, which repeats the work of the 1928 Great Barrier Reef Expedition. Our findings sadly highlight the pressure and future of reefs in Australia and elsewhere. For us, this study was truly an inspirational experience, to update the study done by some of the pioneers of modern marine biology.

We are certain that this study is one of many that are likely to quantitatively revisit the findings of the 1928 expedition.

We have modified the title of the manuscript to read "**Ecological changes over 90 years at Low Isles on the Great Barrier Reef**", in accordance with Reviewer 1 and in order to be consistent with Hamylton et al. (2019) which deals with geomorphological changes at Low Isles. Our specific responses to the three reviewers' very useful comments and suggestions are outlined here. **[Now, with line numbers, indicating where changes were made].**

Reviewer #1:

Fine et al. is a sound and novel assessment of a long-term ecological data set on a coral reef- indeed the longest such data set that there is. The paper will be useful to scientists in the field for its consideration of such a long-term data set, consideration of coral and reef health indicators beyond just coral cover, and comparisons between drivers of change at different temporal and spatial scales (e.g. water quality vs cyclones). I also like the incorporation of long-term physical and chemical information with the biological.

My only real concern relates to replication and generalization. Given the locations of the three 'traverses' they don't look like replicate samples of the same habitats at LI reefs, rather each is a 'traverse' through a different habitat within the Reef system. Of course, this isn't under the control of the authors, and I'm not expecting them to redesign surveys conducted in 1928. But I do think the paper should give consideration to what habitats were sampled and to what extent there is any replication. Does the survey design allow generalization about LI reef as a whole, or just the three specific 'traverses'?

We thank Dr. Wachenfeld for his support and constructive criticism. He is very qualified as a reviewer for our study, given his own work on historical sites along the Great Barrier Reef.

The 1928 expedition performed three traverses only at the Low Isles Reef in places and orientations and habitats they found representative. These sites were selected to be representative. In his introduction to the ecological surveys of the 1928 GBRE, T.A. Stephenson explains the rationale for the sites that were selected and describes the habitats. The text plus measurements give us an interesting perspective at how the community is changing across the reef zones.

Had it been done today, one would use replicates over randomized transects, in parallel to depth contours. The GBRE was pioneering in its use of transects (traverses) but were also

limited by diving depth, technology, and wave conditions. Despite these challenges, however, the members of the GBRE took measurements that can be compared with the modern measurements - essentially allowing us to test for whether the measurements from the 1929 expedition could be part of the distribution of measurements undertaken by our team over the past two decades. As we find, the earlier measurements were richer in corals, higher in diversity and lower in terms of soft corals, compared to the last 15 years.

We could only repeat their traverses, which all start on the reef flat and continue towards the edge of reef flat into the reef slope and to the sandy bottom. Because it was performed at different orientations of the reef, the length of the traverses could be shorter or longer depending on the steepness/extension of the subtidal section but they all go through similar habitats with orientation-dependent (wave action currents etc.) changes. We believe, consequently, that the 3 traverses are characteristic (given the logistics back then) of the low Isles Reef.

It is also important to note that the AIMS Long Term Monitoring program surveys around this reef are also in agreement with the recent measurements undertaken by our team. These surveys still use very similar techniques for their permanent transects, and add on top Manta-tow surveys for a general impression of reef state.

We repeated each of the Traverses three times (see methods) in 2004, 2015 and 2019 to increase our accuracy. Furthermore, we examined selected sites in the intertidal region – comparing them to the 1928 and 1954 surveys. We are confident that the surveys performed in 1928, 1954 and in the present study are a good representation of the LI reef.

This concern is amplified when I consider the title's reference to 'the northern Great Barrier Reef'. It is debatable whether or not the three 'traverses' are representative of LI reef as a whole, but I contend they are almost certainly not representative of the northern GBR as a whole. LI reef is close to shore adjacent to a part of the catchment with substantial agricultural development. There are other northern GBR reefs in similar locations, but most of them are quite different (far offshore and/or not adjacent to intensively developed catchment). I think the title should be edited to better reflect the true geographic scope of the data presented and the text should give some consideration to the relationship between LI reef and other GBR reefs. This is a great paper and I look forward to seeing it published.

Overall, I th

The reviewer asks an important question. How representative is the Low Isles of the northern GBR? Clearly, the novelty of our long-term study rules out comparison to parallel studies of equal length. However, several other studies of northern Great Barrier Reef reefs report similar trends for the last few decades. Lam et al. 2018 studied inshore reefs between 2004 and 2015 and show that 46% of the 15 studied reefs declined in coral cover, 23% fluctuated. For Low Isles we show that it does indeed fluctuate within decadal scales but show an overall declining trend over the last century, highlighting the importance of long-term perspective. We fully agree that we should be cautious in generalizing our results and conclusions. Hence, we suggest a modification of title of the manuscript to: "**Ecological changes over 90 years at Low Isles on the Great Barrier Reef.**", [Line 1] We have also added a paragraph on the issues associated with generalizing the study [Lines 190-198].

Reviewer #2:

General Comments

The authors have presented a unique set of data on a coral reef that spans 90 years. There is great concern globally with respect to how natural systems are changing. This paper will be well received as there are few base lines, especially for coral reefs, to go from. Furthermore the 1928-29 expedition collected both physical and ecological data which adds value to the long-term comparisons. Other positive points – the data are from the intertidal and subtidal, multiple taxa were studied including corals, molluscs, echinoderms macro crustaceans and reef architecture. It is interesting that one of the ‘villians’ damaging reefs today, the Crown-of-thorns star fish, has disappeared. Not only was coral cover studied, but changes in species composition (e.g., the replacement of branching corals with massive corals) were documented. Again few studies have been encompassed such a long period to document such changes.

Thank you, we appreciate the support for our study .

See ‘Minor comment re Fig 1 – the drop in pH seems strangely linear! True, this illustration was aimed at highlighting changes in environmental conditions since 1928, not claiming to give the exact values for the entire period. We added a statement to the figure legend to explain this and changed the pH curve to reflect a non-linear change over time.

Minor Comments

Line 19 – it is unclear how many cyclones have actually caused damage?

This data is available from Supplementary Table 3, based on anecdotal reports of damage to the reef, the LI or Port-Douglas

Lines 32-54 – the ‘Main section/Introduction is coherent and clearly states why there would be international interest .

Thanks

Line 83 re reference 10 – the authors should provide some mainstream literature on this temperature change – certainly CSIRO has published changes from the 1960s

Huang et al. was replaced with Lough et al. 2018 (ref 12), "Increasing thermal stress for tropical coral reefs: 1871-2017" which has the supporting data in its supplementary material [**Line 292**].

Line 90 – another good reference on runoff and nutrient levels is Furnas (2003) ‘Catchments and Corals: terrestrial runoff on the GBR

Thanks for that-Furnas (2003) added [**ref 14, Line 297**].

Line 91 – the measured rise in sea level is interesting. You would expect the intertidal to be more protected with more water, but that does not appear to be the case? Are there data on the growth/erosion of shallow carbonates.

Perry et al. 2018 (Nature) suggest that coral cover strongly predicts the capacity of reefs to track sea level rise (SLR). Therefore, for the Low Isles, under the current low coral cover, SLR might

result in 'reef drowning'. It also depends on the rate of SLR as shown by Hamylton et al. 2014 for Lizard Island [Line 131-133].

Line 112 re: 'near local extinction – of course this refers to the sampling sites – not necessarily the whole reef which may have refugia?

This is true and we have clarified this in the text to reflect that it is near extinction of some species in the habitats studied where they were common in 1928 and persisted until at least 1954 [Line 99].

Line 145 the number of intertidal sites (n=10) from 1954 was convincing.

Yes, (12 in fact), we were convinced that it is representative and striking evidence of change.

Line 148 – re loss of other invertebrates (e.g., echinoderms, molluscs crustaceans), there was no comment on why - loss of food such as live coral, destruction of reef architecture. etc?

We tend to think the loss of species is a consequence of losing reef architecture (i.e. habitat) although it is likely to be a combination of direct factors such as changing sediment/nutrient effects, or physical change and the loss of habitat. Now discussed in the text [Lines 149-153].

Line 164 Re: species of soft coral – that is a species diverse group – are any species or family names available? Good point, the genus level of major players is now mentioned in the text where possible [Line 114] (e.g. Sarcophyton, Lobophytum and Sinularia which dominate the area. We reflect the key groups to match that of earlier studies (i.e. going to finer levels of taxonomy isn't required).

Figures – the quality of figures could be improved – perhaps redrawn (e.g. Fig. 1). I suggest the addition of a present day google map would be complementary – there is space below 1d!? Figure one has undergone redrafting to improve quality. Fig. 1 A, was improved to represent better the section of the GBR where the study was performed. Fig. 1C will now be (Fig. 2) so that it can be presented Larger. Figure 2a (now 3a) was improved in accordance with Rev 3 comments.

Figure two – there are no error bars for 1928 – say why there was only one value in the records?

In 1928-9 three traverses were performed but each one is in a different orientation of the reef. Therefore, we are uncomfortable with respect to adding error bars to this data. In the present study we repeated three times each traverse therefore error bars could be added [Methods, Line 219 onwards].

Live cover does not distinguish hard coral from soft coral? - I suggest the symbols are linked with lines as that would make the long-term trends easier to interpret.

As stated in the figure legends Fig. 3a is live hard coral cover in traverse T1-3.

We linked the symbols with lines as suggested and indeed, it represents the fluctuating coral cover much effectively.

Figure 3 it is hard to be convinced about some of the comparisons given the image quality. Perhaps larger paired image in the extended figures would be helpful.

Agree, we believe the present form of the figures is much clearer.

Re comparison c versus d and e versus f. How do you know this was the exact same location? GPS targeting may not be good enough for this scale.

According to official US government information about the Global Positioning System, GPS accuracy of a smartphone is about 5 m and more dedicated devices are in the few centimeters range. We used dedicated GPS to get to sites and ground-proved its accuracy. As discussed in Hamylton (2017 and 2019), the 1928 mapping was highly accurate and following georeferencing this map we were able to return to the same spot. This was tested using existing places since 1928 such as the lighthouse, base of the old Jetty etc. some features on the reef are very obvious when you arrive at the spot (the bolder tract for example, Luana and Wisharts reefs etc.) [See Methods, Line 207].

Re: extended figure 1 – it would be good to see percentage cover by depth. This is a more common metric in the literature. Further colony number may not relate to percentage cover very well – eg one colony may cover a large area while small colonies may be abundant, but cover little.

Extended Figure 1 (now Supplementary Fig 1) is built on a figure produced by Manton in the GBR expedition and published in Spender 1930. Our intention was to use the same data representation with data from our study. We agree with the reviewer that it will be interesting to look at coral cover along a depth gradient but, (1) we are limited to the data the 1928 expedition collected so hard to compare and (2), we find that coral cover fluctuated so much that it is not a good predictor of the reef state [see discussion beginning in Line 169].

The extended tables add value to the paper.

Noted.

Reviewer #3:

This is a very interesting paper about long-term trends in reefs community dynamics. The study shows that communities have experienced major phase-shifts over the last 90 years, with few signs of return to the initial state. Specifically, massive corals have replaced branching corals, and soft corals have become much more numerous than hard corals. Others invertebrates, such as grazing echinoderms, gastropods, crustaceans and many other non-coral invertebrates are also disappeared. 1) Authors consider multiple factors to explain these changes, however, they don't specify how exactly each one could explain the changes observed in the community.

The intertidal zone is the most temporally and spatially variable of all marine habitats. It is characterized by environmental extremes (temperature, salinity, desiccation, nutrient supply), yet there are also strong biotic interactions determining community structure. Therefore, organism living in the intertidal zone have evolved a variety of anatomical, physiological and behavioral adaptations that enable them to survive in this challenging habitat. It seems to me, that recurrent storms could explain by itself most of the changes, and the fact that subtidal coral communities, in contrast to intertidal coral communities,

did not show consistent decline over the past century seems to support this hypothesis, but I would like the authors to delve deeper into how others different factors considered could explain the changes registered: sea level rise, for example, must be also very important. In the time that the authors made the study, the original intertidal zone is not so intertidal, because the sea rose over 15 cm (see Fig. 1).

We have now added to the discussion on the effects of multiple stressors on coral assemblages within the restriction of the available space [Line 129 for modification of ramparts, 133 for sea level rise, Line 136 for cyclones, Line 158 for multiple acute and chronic disturbances].

Given the tidal range in the area and the fact that the reef flat still gets exposed during spring low tides, it is still considered intertidal but maybe not as extreme.

2) They also may consider other abiotic factors that could be responsible of the observed data, such as wave exposure and tidal height, which have been shown to have strong effects on the dynamics of intertidal communities, contributing to invertebrate abundance and species richness; all this is also related to the rise in sea level.

We agree with the reviewer's comments and hence wave action is discussed as part of the influence of cyclones [paragraph beginning in Line 136], and that may be a factor in the reef side facing the cyclones. We have no evidence of changes in tidal range but sea level did change (higher today by about 20 cm compared with 1928). This however should reduce the stress caused by long exposures during low tide periods (assuming that the reef has not kept pace with sea level rise, which is unlikely [see discussion beginning in Line 132].

3) I also would like that authors explore additional hypothesis, which serves to explain the impoverished diversity, based on the unstable composition of faunas in remote marginal regions, and in the low resilience of these faunas (see references below). Due mainly to physical perturbations commented above, species already living near their tolerance limits become locally extinct and are not soon replenished after disturbances (strong cyclones) because of their isolation from source populations.

Interesting point but in fact, Low Isles is not so remote or isolated. As an inshore reef it is surrounded by many reefs and the mainland (less than 10 Km to Snapper Island, about 14 Km to Toung and Bat reefs), potential source reefs are very close. There is no reason to believe that it is the supply of propagules but rather their survival (due to repetitive disturbances) that results in impoverished reefs around Low Isles [discussed in Line 155 and 175 onwards].

4) Apart from corals, invertebrates that were reported to be common in these sites were no longer present, despite extensive searches across a substantial area of intertidal. Could be more specific pls? Species? Genera?

please see Supplementary table 2 where we specified which invertebrates were recorded in 1954 and which are there now.

5) Fig. 1: Pls define CoTS =

"Crown of Thorns Starfish" has been added [Lines 89 and 407].

6) Surveys of the intertidal sites described by the GBRE and those of a later 1954 GBR expedition, suggest coral cover has declined along with a major decline in species

richness. However, coverage seems to increase in T3 (Fig. 2). Coral cover in 2015 was high in T3 (53%), was low in all three traverses in March 2019. Could explain a little more why cover oscillated, and even increased, and in contrast species richness consistently declined over time?

This has been added to the text **[discussed in Lines 159 onwards]**. Coral cover tends to recover faster, but typically, succession begins with pioneering species occupying space. Only later, with intermediate disturbances, other, less competitive species are able to penetrate the community towards buildup of a rich climax community. In the case of the Low Isles, following disturbances, pioneering species **[Line 164]** do occupy available space but are not joined by other species, or at least very slowly.

7) Please consider these relevant references in the revision of the ms

Thanks to the reviewers. We have added three out of the 4 references. Note that the paper by Garry Russ deals with fishing in marine reserves and hence is not as relevant **[refs 27-29]**.

REVIEWERS' COMMENTS:

Reviewer #1 (Remarks to the Author):

I'm happy with the revised paper and recommend publication